# Zinc Fortification: Current Trends and Strategies

**DOI:** 10.3390/nu14193895

**Published:** 2022-09-21

**Authors:** Andrew G. Hall, Janet C. King

**Affiliations:** 1Department of Nutritional Sciences & Toxicology, University of California, Berkeley, CA 94720, USA; 2Department of Nutrition, University of California, Davis, CA 95616, USA

**Keywords:** zinc, nutrition, fortification, biofortification, bioavailability, global health

## Abstract

Zinc, through its structural and cofactor roles, affects a broad range of critical physiological functions, including growth, metabolism, immune and neurological functions. Zinc deficiency is widespread among populations around the world, and it may, therefore, underlie much of the global burden of malnutrition. Current zinc fortification strategies include biofortification and fortification with zinc salts with a primary focus on staple foods, such as wheat or rice and their products. However, zinc fortification presents unique challenges. Due to the influences of phytate and protein on zinc absorption, successful zinc fortification strategies should consider the impact on zinc bioavailability in the whole diet. When zinc is absorbed with food, shifts in plasma zinc concentrations are minor. However, co-absorbing zinc with food may preferentially direct zinc to cellular compartments where zinc-dependent metabolic processes primarily occur. Although the current lack of sensitive biomarkers of zinc nutritional status reduces the capacity to assess the impact of fortifying foods with zinc, new approaches for assessing zinc utilization are increasing. In this article, we review the tools available for assessing bioavailable zinc, approaches for evaluating the zinc nutritional status of populations consuming zinc fortified foods, and recent trends in fortification strategies to increase zinc absorption.

## 1. Introduction

Zinc, an essential trace element, has a broad range of critical biological functions that span all life stages and essential functions, including reproduction, growth, metabolism, neurological, and immune functions [1]. Zinc deficiency is widespread among populations around the world and, thus, underlies much of the global burden of malnutrition. Based on the prevalence of stunting, a symptom of zinc deficiency, among children under 5 years of age, or the prevalence of low plasma zinc or low zinc intakes, zinc deficiency is the most common nutritional problem worldwide [2]. Zinc deficiency is of particular concern among infants, children, and women of reproductive age, although it also occurs in adolescents and older adults. The prevalence of zinc deficiency is also higher in rural than urban areas, although it is present in both.

The intake of staples is also higher in rural over urban areas. Thus, zinc fortification strategies, i.e., fortification or biofortification of staple foods such as wheat, rice, or maize, will likely improve the zinc intakes of vulnerable populations worldwide. Although zinc fortification efforts have advanced recently, optimizing the amount of zinc absorbed and monitoring the health impacts due to fortification remain significant challenges. Both aspects are linked to the bioavailability of zinc, i.e., the proportion of zinc in food items that is released during the digestive processes, is absorbed, and is utilized for numerous biological functions [3].

Strategies for designing zinc fortification programs have aptly emphasized the importance of considering zinc bioavailability [4,5,6]. Zinc fortification research studies tying the design and monitoring of zinc fortification programs to advances in our knowledge of the influence of dietary composition on zinc absorption and retention in tissues, and utilization in cells, are reviewed. Finally, specific strategies for future programs to increase the impacts of zinc fortification are discussed.

## 2. Dietary Factors Influencing Zinc Absorption and Retention

The amount of zinc absorbed is frequently used to evaluate the impact of zinc fortification strategies. Zinc absorption generally increases with the amount of dietary zinc, and is strongly influenced by the composition of the overall diet. Phytate, the storage form of phosphorus in plants, reduces zinc absorption [7,8]. Phytate binds zinc ions with high affinity [9] and sequesters zinc from the active transport mechanisms located on the surfaces of cells in the lumen of the small intestine.

In grains, phytate is co-located with the zinc and protein-rich bran and germ [10]. Phytate is also high in legumes, seeds, and nuts, other plant sources of protein and zinc [11]. In contrast to phytate, dietary protein itself enhances zinc absorption, independent of the protein source [8]. Animal source foods are rich in protein and lack phytate. Thus, adding animal sources of protein readily increases the fractional zinc absorption from the whole diet without interference from phytate. Plant sources of protein, in contrast, typically also have the highest phytate content [11]. Although adding plant protein also increases zinc absorption, this effect may be countered if phytate is not reduced.

Dietary phytate and protein also influence the retention of absorbed zinc. When a meal is consumed by adult men, 2.4 to 4.8 mg of zinc from tissues are immediately secreted into the digestive tract with digestive juices [12]. This secreted zinc is reabsorbed during the passage of ingested food through the small intestine. The plasma kinetic response to orally ingested zinc demonstrates an enterohepatic recirculation [13,14].

The amount of zinc secreted during the digestive processes is substantial. It is equivalent to the amount of zinc that must be absorbed from diet each day in order to meet physiological requirements [15]. Just as digestive enzymes are efficiently conserved via reabsorption [16], the co-absorption of zinc secreted with digestive enzymes and peptides released into the gut postprandially assures the reabsorption of digestive zinc [17] and, thus, conserves body zinc. However, dietary phytate also binds with the tissue zinc secreted endogenously into the small intestine following a meal, and impairs its reabsorption [18].

## 3. Techniques for Estimating the Impact of Zinc Fortification on Zinc Absorption

Direct measurement of fractional zinc absorption using the dual isotope tracer ratio (DITR) method is the most reliable technique for measuring zinc absorption (Figure 1), and is used extensively in the context of zinc fortification [19]. For example, the DITR method was recently used to compare the absorption of different zinc salts used in post-harvest fortification [20,21], for determining the interaction with other mineral fortificants such as iron [20], and for comparing different fortification modes, such as biofortification versus post-harvest [22,23], or fortified maize versus fortified water [24]. When the studies are done in local settings where fortification will be implemented, the DITR results can be used to evaluate the potential impact of fortification strategies towards meeting a population’s zinc physiological requirements [21,25,26,27,28,29,30].

Since the dietary intake of both protein and phytate influence zinc absorption, the composition of the usual diet should be considered when designing zinc fortification strategies towards meeting physiological requirements. Zinc stable isotope methods may readily be applied to test zinc fortified foods in the context of meals containing local, commonly consumed foods or feeding an overall dietary pattern typical of that consumed locally. Recent examples of this approach have been published [25,26,27,28].

Disadvantages of zinc stable isotope tracer techniques for assessing zinc bioavailability include the cost and subject burden. These disadvantages limit the number of studies available to assess the efficacy of providing zinc fortified foods to various populations. A less expensive approach is to measure the composition of typical diets consumed by the target population, including dietary components likely to affect zinc absorption. One example is the comparison of the phytate to zinc molar ratio [31]. However, since zinc absorption is non-linear with respect to the total dietary zinc, the simple evaluation of the phytate to zinc molar ratio alone does not produce a reliable estimate of zinc absorption from the whole diet.

Miller and colleagues modeled zinc absorption based on the dietary phytate and zinc intakes from 105 subjects in 11 zinc tracer studies and produced equations that more reliably predict zinc absorption in adults based on daily zinc and phytate intakes [7]. Since several studies found that dietary protein, calcium, and iron also influence zinc absorption, the original model was fitted to data from 43 subjects that included intakes of those nutrients as well as zinc [8]. The resulting model showed significant effects of dietary protein and calcium, but not iron, on zinc absorption.

A similar approach was used to develop a model of zinc absorption in children with a mean age of 24 months [32]. The model showed that child age was the only predictor of zinc absorption besides the dietary zinc content. Possibly, the lack of a significant phytate effect was due, in part, to insufficient statistical power. Few studies included phytate intake data, and only one of the eleven studies included in the model varied phytate intakes to determine its effect on zinc absorption in children. The authors of that study reported that reducing the phytate content in a corn and soy porridge improved the zinc absorption of 3- to 13-year-old children recovering from tuberculosis, but not in control children [33]. Furthermore, the effects of various protein intakes, and of breastfeeding, were not evaluated. Pureed beef was the only non-fortified complementary food evaluated that increased zinc absorption sufficiently to meet physiological requirements [34].

In contrast, two more recent zinc tracer studies demonstrated a strong effect of phytate reduction on zinc absorption among young children [29,30]. When the phytate was degraded enzymatically, zinc absorption from a zinc-fortified millet porridge increased 68%, from 9.5% to 16%, among Burkinabe children 12 to 24 months of age [29]. In Gambian children aged 18 to 23 months, similarly reducing the phytate content of a millet porridge that was fortified with a lipid-based micronutrient supplement, increased zinc absorption by 86%, from 8.6% to 16% [30]. These studies indicate that more data on phytate and zinc absorption among children are needed, towards developing models that could be the basis for planning fortification strategies.

Modeling zinc absorption is a useful approach towards the optimization of a fortification strategy for a particular food [35,36,37]. For example, modeling enabled the evaluation of wheat varieties for zinc biofortification in Pakistan [35]. A pool of 65 wheat varieties under consideration for genotype development were compared. Depending on the zinc and phytate content of each variety, it was estimated that 1.5 to 2.2 mg of zinc would be absorbed from a typical daily intake of 300 g of wheat. A similar modeling approach was used to optimize the amount of zinc added to the soil for agronomic fortification of wheat [36], and the foliar application of zinc to wheat [37].

Since zinc absorption from a staple such as wheat is dependent on the composition of the rest of the diet, modeling studies that include data on the usual diet would further allow comparison of the potential impacts on absorbed zinc relative to the physiological requirements. These applications of modeling only require the additional determination of usual zinc and phytate intakes in the population of interest [7], and may further be enhanced with inclusion of protein and calcium intakes [8]. They thus represent a cost-effective approach towards comparing the potential impacts of fortification strategies on the prevalence of absorbed zinc below the physiological requirements, before more detailed zinc tracer studies are done.

Joy and colleagues modeled zinc absorption to estimate the potential impact of HarvestPlus targets for fortifying staple crops with zinc in Africa [38]. The authors used Food Balance Sheets for 46 countries that were integrated with food composition data to estimate the per capita intake of zinc and phytate. The mean risk of zinc deficiency, estimated as the proportion of intakes below the estimated average requirement, was 40%. If the HarvestPlus conventional breeding targets for rice, wheat, maize, pearl millet, beans, cassava, and sweet potato were met, the estimated risk of zinc deficiency would be 4%, representing a 90% decrease.

More recently, this modeling approach was included in the evaluation of a community-based fortification program in Cameroon, in place of tracer studies [39]. The changes in absorbed zinc were predicted using Miller’s 2007 equation and compared with the physiological requirement for absorbed zinc. Another study showed that fortifying biscuits with zinc would not substantially change the absorbed zinc, given the amounts of zinc fortified wheat flour already consumed in the usual diet [40].

An updated version of Miller’s model includes additional dietary factors affecting zinc absorption [8]. When adding protein and calcium to the original model of phytate and zinc, the proportion of variation in zinc absorption explained by the model increased from 82% to 88%. Furthermore, the model demonstrates how zinc fortification may be effectively combined with other strategies such as phytate reduction and increased protein intake to optimize zinc absorption. However, we are not aware of studies using Miller’s updated model to evaluate impacts of zinc fortification on the risk of zinc deficiency.

Another useful method for comparing the luminal availability of zinc from fortified foods uses in vitro digestion coupled with a cell model of immortalized human small intestine enterocytes, Caco-2 cells [41]. This approach is relatively inexpensive and correlates with zinc absorption in animal models. To our knowledge it has not been compared with direct measures in humans. The method has recently been used for evaluating zinc biofortification of rice [41], phytate reduction of sorghum [42], beans targeted for zinc biofortification [43], food-to-food fortification in combination with zinc fortified maize [44], and zinc biofortified low phytate rice [45].

These studies represent an increasing trend towards use of Caco-2 cell models to predict zinc bioavailability. Since this method measures zinc uptake by cultured enterocytes, and not zinc absorption across the lumen into plasma, data from this approach best indicate luminal availability, or the proportion of zinc in the food matrix released by digestive processes and available for transport into the intestinal epithelium. Given the potential differences in digestion and cellular zinc uptake in vitro vs. in vivo, and multiple whole-organism factors that may influence zinc availability in the lumen, the future use of this technique to estimate zinc bioavailability should be supported by validation with zinc tracer studies in humans.

## 4. Zinc Utilization and Trends in Zinc Biomarkers for Monitoring Fortification

Bioavailability includes the proportion of a nutrient in diet that is ultimately retained and used in the body [3]. While dietary factors affecting zinc availability in the lumen and zinc absorption stem from decades of research, data connecting these factors to tissue zinc uptake and utilization in zinc-dependent functions are still limited. This is due in part to the lack of zinc-specific biomarkers responsive to small but biologically significant changes in zinc intake. The efficacy and effectiveness of zinc fortification efforts in low- and middle-income countries thus remain uncertain.

The most commonly used biomarker of zinc status, serum or plasma zinc concentration, naturally fluctuates within individuals by 18% to 22% over a 24-h period [46,47]. This is similar in magnitude to the peak response in plasma zinc concentration following daily zinc supplements taken in the fasted state [48]. However, zinc taken with food, as is the case in fortification, exerts a diminished plasma response compared with zinc taken in the fasted state [49,50]. Furthermore, when the dietary zinc supply is low, the body avidly conserves zinc by reducing endogenous losses [51] and, therefore, sustains plasma zinc levels. Consistent with the physiological need to conserve circulating zinc over a wide range of zinc intakes, a recent review of NHANES dietary data demonstrated the lack of a correlation between zinc intake and serum zinc concentration [52].

The high variability of plasma zinc concentration within individuals, coupled with the physiological priority to maintain plasma zinc concentration when zinc-dependent functions in tissues may begin to decrease, limits the usefulness of plasma zinc as a biomarker for changes in zinc intake or utilization. Identifying alternative biomarkers for detecting a suboptimal zinc status, towards monitoring the effects of zinc interventions, is challenging due to the diversity of zinc functions and limited specificity for zinc. The clinical manifestations of zinc deficiency, such as impaired growth or immune function, are not unique to zinc. Nonetheless, biomarkers of zinc function may provide valuable information on important changes in zinc utilization in response to fortification.

Several studies have demonstrated that changes in health outcomes or related biomarkers reflect changes in zinc utilization in response to changes in zinc intake, without corresponding changes in serum or plasma zinc [26,53,54,55,56]. The earliest functional measures of zinc status are related to child growth and morbidity due to infections [1]. However, changes in growth may also be due to changes in energy intake or infections. This contributes to their variability, and these endpoints do not exhibit a clear response to zinc fortification [57,58]. Recently, functional biomarkers of zinc, responsive to the small changes in dietary zinc intake that are typical in zinc fortification, have been identified. Some of these biomarkers, including circulating fatty acids, DNA strand breaks, biomarkers of immune function, zinc transporter expression, and hair zinc, have potential in monitoring zinc fortification effects.

Plasma fatty acid concentrations, and estimates of fatty acid desaturase activities, based on the ratios of product to precursor omega-6 fatty acids, have been identified as novel biomarkers of zinc status due to a strong correlation with plasma zinc concentrations [59,60]. Additionally, zinc may be more efficiently directed towards fatty acid desaturation when taken with food compared with zinc taken in the fasted state [49]. Although their use as zinc biomarkers is of further interest due to their relation to cardiometabolic health, more studies are needed to determine the potential usefulness of fatty acids and related metabolites for monitoring the effects of zinc fortification.

Zinc also has multiple roles for preventing DNA damage and supporting DNA repair mechanisms [61]. DNA strand breaks, measured by the comet assay, increased following 2 weeks of dietary zinc depletion and recovered following 4 weeks of consuming a modest 10 mg zinc per day [54]. In Ethiopian women, 17 days of a 20 mg zinc supplement reduced DNA strand breaks compared with a placebo control [56]. Neither study detected a change in plasma zinc concentration. These data suggest the comet assay may be a promising biomarker for monitoring the effects of zinc fortification. However, no randomized trials have assessed the response of DNA strand breaks to zinc fortification [58].

Recent studies demonstrated that biomarkers of immune function may also be more sensitive than plasma zinc concentration, to changes in zinc intake due to fortification. Costarelli and colleagues examined the effects of zinc-fortified skim milk providing 4 mg zinc per day for 60 days in a study of elderly adults, compared to control milk with no added zinc [62]. This small increase in dietary zinc increased the levels of cytokines associated with cell-mediated immunity, anti-inflammatory cytokines in peripheral blood mononuclear cells, and plasma thymulin activity, without corresponding increases in plasma zinc concentration.

The measurement of zinc transporters should also be further studied in the context of zinc fortification. Animal studies show compensatory changes in zinc transporter expression in response to relatively small differences in dietary zinc intake [63]. The provision of 6.6 mg zinc per day for 27 days from fortified milk to adolescent girls increased zinc importer ZIP1 mRNA levels in peripheral blood mononuclear cells, also without corresponding changes plasma zinc concentration [64].

Since the amount of zinc deposited in hair during follicular growth is sensitive to changes in zinc intake, the response of hair zinc content to intake of zinc fortified breakfast cereal was studied in children over 40 years ago [65]. However, recent randomized studies have not examined the effect of zinc fortification on hair zinc content [66].

Zinc kinetics is another area that has been explored for assessing the impact of zinc fortification. In a study of home fortification of a rice and lentil-based complementary food, the addition of only 10 mg zinc per day increased the size of the exchangeable zinc pool (EZP), i.e., a cellular zinc pool that exchanges most rapidly with plasma [26]. However, no significant changes in EZP occurred when zinc absorption was increased by reducing dietary phytate [30].

Although the number of studies is limited, the results support the usefulness of zinc functional biomarkers reflecting changes in cellular zinc uptake and utilization in response to fortification. Since small changes in dietary zinc intake typical of fortification programs affect zinc functional biomarkers more readily than plasma zinc concentrations, future studies of zinc fortification would be a suitable forefront for developing novel zinc biomarkers.

## 5. Trends in Zinc Fortification Strategies

Fortification of staple grains post-harvest by the addition of inorganic zinc salts, typically zinc oxide or zinc sulfate, is the traditional approach to zinc fortification. Post-harvest fortification is often used in flours produced from staple grains. While the technique is well-established, it has several limitations. It requires sustaining well-controlled centralized processing to ensure homogenous distribution in the flours. This represents a major component of the post-harvest fortification expense (Table 1). In many settings where the risk of zinc deficiency is high, the availability of resources limits the implementation, coverage, and sustained use of post-harvest fortification strategies.

Adverse effects may also occur if excessive zinc is inadvertently added to the target food, or pockets of high zinc levels exist due to uneven distribution during the fortification process. Unlike most other nutrients, there is a narrow window between zinc recommended intakes and the upper limit of intake before adverse effects are observed. While recommended daily intakes for adults typically range from 8 to 19 mg zinc per day, the upper limit is only 35 to 45 mg [15], i.e., a 2 to 4-fold increase. Zinc upper limits are based on reduced activity of the antioxidant protein, copper-zinc superoxide dismutase, in erythrocytes [67]. However, women supplemented with a modest 22 mg of zinc daily for 6 weeks, about half of the upper limit, had detectable increases in zinc protoporphyrin due to the incorporation of zinc in place of iron into hemoglobin during erythropoiesis [68].

Furthermore, due to the low solubility of zinc oxide at a neutral pH [9], variability in stomach acidity may determine how well the zinc is solubilized and absorbed. Zinc from porridges fortified with zinc oxide has a lower fractional absorption than zinc sulfate [21]. Zinc oxide, when given as a supplement, also has a lower fractional absorption than zinc gluconate or zinc citrate [69]. Recent studies in animals further showed negative effects on gut health, including increased oxidative stress and a shift in the microbiome towards a genetic profile consistent with increased antibiotic resistance, when zinc sulfate or zinc oxide fortifying salts were compared with zinc with amino acids complexes [70,71].

Soluble complexes of zinc with amino acids or zinc-chelating peptides from protein hydrolysates, are an alternative to inorganic zinc salts. These zinc complexes appear to enhance zinc uptake into cultured cells, and zinc absorption in animal models [17,72,73,74]. In humans, the response in serum zinc concentration over a six-hour period following a dose of 20 mg zinc as a zinc histidine complex was greater than the same amount as zinc sulfate [75]. Similarly, the serum response over the course of 8 h following a 15 mg dose of zinc as zinc bisglycinate was greater than the same amount as zinc gluconate [76]. Further study is indicated to evaluate the potential of zinc amino acid complexes for zinc delivery in food fortification.

Biofortification, accomplished through selective breeding, transgenic crops, and/or agronomic fortification by adding zinc to the soil or to growing crops, increases the intrinsic zinc content of staple foods [77]. There is an increasing trend towards zinc biofortification as a more cost-effective and sustainable option for increasing zinc intakes. Biofortification has several advantages. Crops that are not typically eaten as flours, such as rice or beans, are more readily fortified using biofortification because zinc is incorporated into the edible portion of the plant [78]. Compared with post-harvest fortification, the lack of a need for special processing reduces the cost of biofortification after the initial crop development. Recent crops targeted for zinc biofortification include wheat [36,79,80], rice [60,81,82], maize [27,83], pearl millet [25,84], beans [43,85], and bananas [86].

Biofortification targets proposed for use in Africa could potentially reduce the risk of zinc deficiency 10-fold [38]. Although there are physiological limits to the amount of zinc that may be incorporated into crops through biofortification [87], excessive or uneven distribution of zinc within the crop is unlikely. Furthermore, zinc from biofortified crops is well-absorbed. Fractional zinc absorption did not differ between biofortified vs. post-harvest fortified wheat, maize or rice [22,23,27].

Several fortification strategies are more appropriately applied on a small scale, or take advantage of non-staple foods commonly consumed in a subpopulation at particularly high risk for zinc deficiency. Although the fortification of infant formulas and ready-to-eat breakfast cereals to prevent zinc deficiency in infants and young children is not new, these strategies require that zinc is added in the factories where the foods are produced. As with other types of post-harvest fortification, the development of systems for adding and maintaining the appropriate amounts of zinc in the finished food products increases the production cost. Coverage may also be limited by market forces or poor distribution within a population. Alternatively, home fortification packets that are readily produced, stored, and distributed, may easily be added to complementary foods at home. Consequently, recent zinc fortification strategies targeting infants have included the home fortification of complementary foods with zinc [26,50,88,89,90].

In a study of Kenyan infants, the provision of 5 mg zinc per day increased zinc absorption enough to meet the physiological requirement, and it increased the EZP compared with control [90]. However, in Cameroon, the provision of 6 mg zinc/day in young children failed to alter plasma zinc concentrations [50]. Higher amounts of zinc for longer periods of time may be needed to change zinc biomarkers. For example, home-fortified complementary foods providing an additional 10 mg zinc per day did not increase the EZP after 3 months, although extension of the program to 9 months did [26]. Similar goals may also be achieved through biofortification. Increasing the zinc intakes by 2.7 mg/d from biofortified maize in young Zambian children met their physiological requirements for absorbed zinc [27].

These studies further demonstrate that the amounts needed to meet physiological requirements vary. What is adequate for one population may approach the upper limits for another. Daily use of a micronutrient powder that provided 4.1 mg of zinc per packet increased the simulated prevalence of an excessive zinc intake from zero to 50% among young Ethiopian children [88]. Without biomarkers responsive to small changes in zinc intake, it is tempting to further increase zinc intakes in pursuit of changes in zinc biomarkers to support claims of efficacy. However, depending on the needs of the population, this could risk excessive zinc intakes. The challenge is further compounded by limited data regarding the adverse effects of excessive zinc intake in children [91]. Due to the limited knowledge of the safety and efficacy of supplemental zinc in pediatric populations, biomarkers related to adverse effects of excessive zinc intake should be monitored in zinc fortification where current upper limits for zinc are likely to be exceeded.

Recently, there has also been a trend to fortify milk and milk products with zinc for populations other than infants, i.e., toddlers [92], pre-school [93] and school-age children [28,94,95], non-pregnant [96] and pregnant women [97], and the elderly [62]. These studies included the fortification of milk or milk-based beverages [28,62,92,93,94], and yogurt [95,98].

In adolescents, zinc fortified milk providing 6 mg zinc per day increased zinc absorption from 1.1 mg per day to 3.1 mg per day [28]. The researchers used stable isotopic tracers to measure the zinc absorption from the milk and from the rest of the diet at the same time. Interestingly, about 15% of the 2.0 mg per day increase in absorbed zinc was due to the effect of the milk on zinc absorption from the rest of the diet, while 85% of the increase was accounted for by zinc absorption from the milk. These data are consistent with other observations of the effects of milk or milk products on zinc bioavailability from the rest of the diet [99]. The potential of milk or milk products, and dietary protein in general, to increase zinc absorption and retention from the rest of the meal, makes these foods intriguing targets for zinc fortification.

Another recent trend in fortification, food-to-food fortification, is defined as a fortification approach that uses locally available foods containing usefully high quantities of a micronutrient or micronutrients of interest, to fortify another food [100]. Since the flavor and consistency of the fortified foods are altered, food-to-food fortification requires recipe development and acceptability testing. While examples are limited, the strategy appears to have good potential for increasing the zinc content of the target foods. For example, gari or tapioca made from cassava may be fortified with soy [101]. This increased the zinc content of tapioca from 0.3 to 1.5 mg zinc per 100 g, and of gari from 0.8 to 1.4 mg zinc per 100 g. Since plant sources of zinc are typically high in phytate, the phytate content should be measured. Concurrent strategies to reduce phytate, (e.g., malting, fermentation, soaking, or germination) should also be developed where feasible.

In contrast, the addition of some foods may partially neutralize the negative effects of phytate on zinc absorption. For example, milk added to a high phytate rice meal increased zinc absorption from the whole meal [96]. The addition of milk or yogurt to a meal of corn tortillas and black beans increased zinc absorption by more than 70%, even though they only increased the zinc content of the meal by 20% [102]. While the addition of dairy products to enhance zinc absorption does not fall precisely within the above definition of food-to-food fortification, these approaches may provide similar or greater benefit increasing zinc absorption.

## 6. Building on Current Trends for Future Progress in Zinc Fortification Efforts

The recent trends in zinc fortification represent a range of creative approaches to address zinc deficiency in diverse settings. The zinc biofortification of locally derived crops, the expansion of the kinds of target foods and zinc sources considered for zinc fortification efforts, and the exploration of new zinc biomarkers linked both to the cellular action of zinc and the etiology of important diseases, are of particular interest. When observing current trends in the context of a more detailed understanding of zinc bioavailability and biomarkers, opportunities to improve and refine new strategies for zinc fortification become more apparent.

In order to optimize their impacts, we propose that future strategies should be based on the following data: dietary intakes compared to predicted impacts on absorbed zinc in relation to physiological requirements, an extended analysis of bioavailability to include the contribution of zinc intakes to zinc-dependent tissue functions, expanded use of functional biomarkers more sensitive to changes in dietary zinc than plasma zinc concentration, and, in large-scale human studies, include an assessment of health outcomes (Figure 2).

Given the influence of phytate and protein intakes on zinc absorption, it is critical that prospective fortification approaches are evaluated and optimized in the context of the foods typically consumed. Specifically, the exploration of potential strategies for zinc fortification should include measurements of the usual foods consumed by the populations of interest. Since phytate intakes may be higher, and protein intakes lower, in settings where zinc deficiency is common, an evaluation of the current intakes of both phytate and protein is necessary. Although updated models for predicting zinc absorption have the capacity to evaluate the influence of the usual intakes of both phytate and protein, recently only the impact of phytate intakes have been considered when developing zinc fortification strategies.

The application of a more rigorous definition of bioavailability further supports the conceptual framework linking zinc intake from fortified foods to its health impacts. In the literature, zinc bioavailability has referred to the following: zinc uptake by cultured intestinal cells following in vitro digestion [72], the appearance of zinc in plasma immediately following an oral dose [75,76], and the fractional absorption of a zinc stable isotopic tracer [20]. While all of these are related to bioavailability, the bioavailability concept should be extended to the utilization of dietary zinc in zinc-dependent processes in the body [3]. For example, with food fortification, where zinc changes are not readily observed in blood plasma zinc concentration, zinc biomarkers that reflect cellular or tissue changes in zinc-dependent activities are important, novel indicators of change in zinc status that need to be developed.

Future efforts should leverage recent advances in zinc biomarker research and develop new indicators for zinc fortification. Potential new biomarkers of the impact of zinc fortification on zinc utilization, for example, include fatty acid metabolites, biomarkers of inflammation and immune function, and of DNA damage. Due to their sensitivity to changes in dietary zinc intake, these functional zinc biomarkers show promise towards evaluating future zinc fortification programs. Although their application to date has been limited, their expanded use may further provide important data linking zinc intakes with the global burden of communicable and noncommunicable diseases.

Zinc deficiency is the most common micronutrient deficiency in the world. Given the lack of biomarkers of an individual’s zinc status, the challenge of linking zinc deficiency to the health burden it causes, must be faced in large-scale population-based studies. Zinc fortification efforts have the capacity to affect changes in zinc intakes on a large scale, and could significantly improve global health. However, many recent fortification studies have not included adequate control that would allow changes in health to be conclusively attributed to changes in zinc intake due to fortification [58].

Compared with iron deficiency, investment in combating zinc deficiency through fortification programs is also likely to be more difficult to establish, due to the limited capacity to assess the impact on the zinc status of vulnerable populations. Nevertheless, since the consequences and economic burden of zinc deficiency are greater than those of iron deficiency, it is essential that improved strategies for assessing the impact of zinc fortification be developed to alleviate zinc deficiency worldwide.

In summary, recent studies illustrate the potential for zinc fortification efforts towards improving the overall health of the world’s populations. Future zinc fortification strategies should integrate knowledge of the current food supply and dietary patterns within different population groups, towards evaluating the predicted effects of the potential fortification efforts on zinc absorption. This optimization should be followed by human studies with expanded zinc biomarkers linking the zinc provided through fortification both to its utilization in tissues and to health outcomes.

## Figures and Tables

**Figure 1 nutrients-14-03895-f001:**
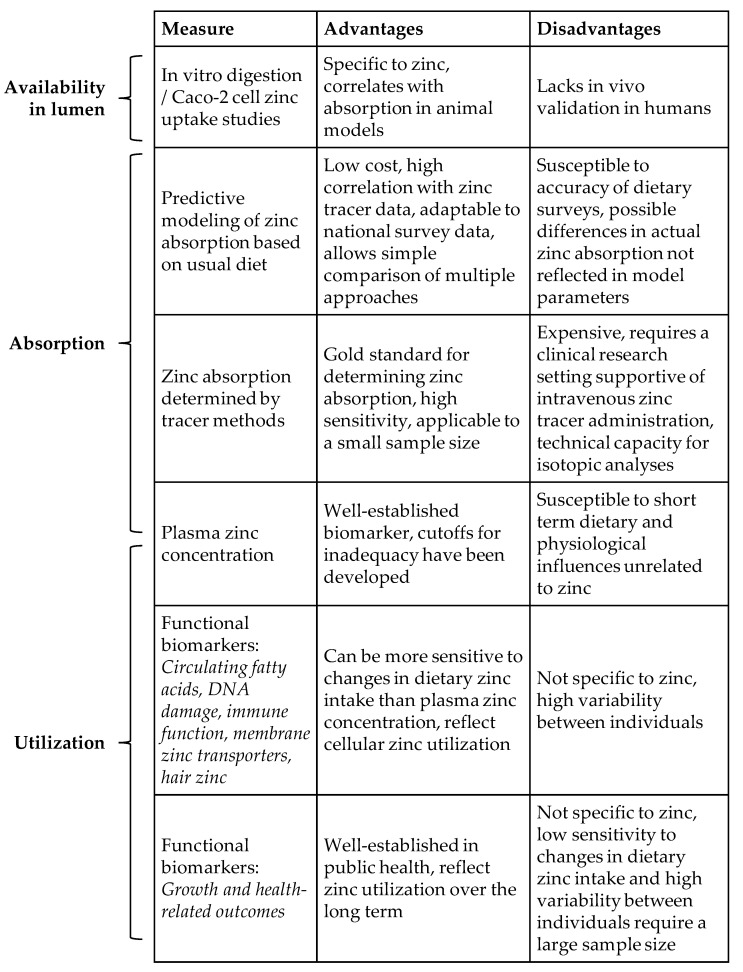
Comparison of zinc measures and relation to bioavailability.

**Figure 2 nutrients-14-03895-f002:**
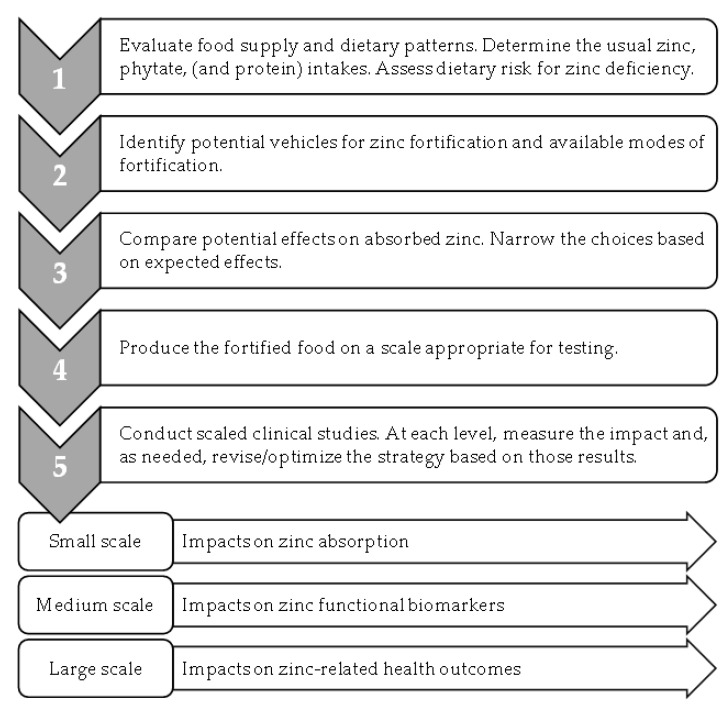
Optimizing zinc fortification efforts.

**Table 1 nutrients-14-03895-t001:** Comparison of recent zinc fortification strategies.

Strategy	Advantages	Disadvantages
Post-harvest fortification of staples	Well-established for use in large-scale fortification of staples that are milled to flours, relatively simple to incorporate at milling	Requires capacity for centralized processing of staples, expensive to maintain coverage and sustained use within a target population, need to control zinc amount and maintain homogeneity, not practical for foods that are not milled into flours (e.g., rice, beans)
Biofortification	Applicable to large-scale fortification of staple crops, no need for special processing, no concern for excessive zinc, practical for crops that are not milled (e.g., rice, beans), low cost of sustained use after initial development	Time and expense of crop development, limitations to the amount of zinc that can be added
Fortification of manufactured food products	Well-established in fortification of population-specific products such as infant formulas or child nutrition biscuits	Production expense, challenges in coverage and sustained use within a target population, need to control zinc amount and maintain homogeneity
Home fortification packets	Readily produced, stored, and distributed, may improve coverage of lower income or otherwise hard to reach populations	Challenges in determining appropriate amount of zinc per packet, opportunity for overuse
Fortification of milk or milk products	High zinc bioavailability, may partially counter inhibitory effects of high phytate diets	Not applicable in populations that do not consume milk or milk products, need to control zinc amount and maintain homogeneity
Food-to-food fortification	Supports small scale implementation in low-resource settings lacking capacity for other modes of fortification	Phytate content of plant sources of zinc can be high, need to test acceptability of resulting flavor and other organoleptic properties

## Data Availability

Not applicable.

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
