# Peer review of "Zinc Fortification: Current Trends and Strategies"

_nutrients, 2022, doi:10.3390/nu14193895_

Round 1

Reviewer 1 Report

The article is well written. However, I feel that this paper can incorporate a lot more technical information than what it currently has at the moment. It is not as thorough as I had hoped. Few more points of improvement:

1) There is extra spacing after periods at several places. Makes for a very distracted reading. 

2) Many key statements are made throughout the paper without references. Please add appropriate references.

3) Need clear transition between lines 34-35. Mention that rural areas consume more staples, etc. 

4) Introduction needs to cover the need for this review and the gaps you are attempting to fill. Should be clear for the reader.

5) Lines 57-60: Need to rephrase. Not all dietary proteins promote zinc absorption. Casein, soy protein inhibit absorption. Plant proteins have high phytate, and hence have two sources of inhibitory compounds, proteins and phytates. 

6) Lines 106-107: Why did iron intakes not show significant influence on zinc absorption? Divalent metal ions compete for absorption in the enterocyte.

7) Lines 116: Remove the word 'or'. 

8) Lines 119-127: What other dietary factors besides phytate are we talking about here? Need to be comprehensive.

9) Lines 156-158: Reformulate.

10) Lines 183-184: Move bioavailability definition to the introduction.

11) Lines 277-278: Isn't zinc absorption regulated at the intestinal level? Under what conditions can one experience zinc overload?

12) Lines 355-357 : This sentence is vague and not needed. Merge with the earlier sentence (353-355). Also, dahi is yogurt. So you can combine references 95 and 98. 

13) Lines 365-367 : The information provided is not true. Clarify that not all dairy proteins promote zinc absorption.  

14) Lines 376: Include at-home strategies to degrade phytate (malting, fermentation, soaking, germination, etc.)

15) Lines 433: Comparison with iron deficiency seems out of place. Reformulate.

Author Response

The article is well written. However, I feel that this paper can incorporate a lot more technical information than what it currently has at the moment. It is not as thorough as I had hoped. Few more points of improvement:

Thank you for your review and comments. We have responded below (blue text).

1) There is extra spacing after periods at several places. Makes for a very distracted reading.

Reply: All double spaces following periods have now been removed.

2) Many key statements are made throughout the paper without references. Please add appropriate references.

Reply: Please provide specific examples of key statements lacking reference.

3) Need clear transition between lines 34-35. Mention that rural areas consume more staples, etc.

Reply: Inserted a transitional statement “The intake of staples is also higher in rural over urban areas.”

4) Introduction needs to cover the need for this review and the gaps you are attempting to fill. Should be clear for the reader.

Reply: The need / gap, and our analysis, is covered in the introduction: “…optimizing the amount of zinc absorbed and monitoring the health impacts due to fortification remain significant challenges,” and, “Zinc fortification research studies tying the design and monitoring of zinc fortification programs to advances in our knowledge of the influence of dietary composition on zinc absorption and retention in tissues, and utilization in cells, are reviewed. Finally, specific strategies for future programs to increase the impacts of zinc fortification are discussed.”

5) Lines 57-60: Need to rephrase. Not all dietary proteins promote zinc absorption. Casein, soy protein inhibit absorption. Plant proteins have high phytate, and hence have two sources of inhibitory compounds, proteins and phytates.

Reply: Rephrased for clarity. We cite evidence from the combined data of available zinc absorption studies that dietary protein itself enhances zinc absorption, independent of the main protein source.  Previous studies that found differences in zinc absorption depending on the protein source were confounded by varying quantities of zinc. In contrast, differences between some isolated proteins, e.g., casein and whey, have been observed (reviewed by Lonnerdal, B. Dietary factors influencing zinc absorption. J Nutr 2000, 130, 1378S-1383S). Why some isolated proteins may have a stronger effect on zinc absorption than others, and how this is dependent on processing, is an interesting and expansive topic beyond the scope of this review.

6) Lines 106-107: Why did iron intakes not show significant influence on zinc absorption? Divalent metal ions compete for absorption in the enterocyte.

Reply: This is covered in the reference cited (8). Further discussion here is beyond the scope of this review.  

7) Lines 116: Remove the word 'or'.

Reply: Replaced with ‘and’.

8) Lines 119-127: What other dietary factors besides phytate are we talking about here? Need to be comprehensive.

Reply: Reworded for clarity, “…more data on phytate and zinc absorption among children are needed…”

9) Lines 156-158: Reformulate.

Reply: Reworded for clarity: “Another study showed that fortifying biscuits with zinc would not substantially change the absorbed zinc given the amounts of zinc fortified wheat flour already consumed in the usual diet [40].”

10) Lines 183-184: Move bioavailability definition to the introduction.

Reply: Reworded. And bioavailability is defined in the introduction.

11) Lines 277-278: Isn't zinc absorption regulated at the intestinal level? Under what conditions can one experience zinc overload?

Reply: In humans, the regulation of whole-body zinc is primarily via endogenous excretion into the intestine. The research supporting the selection of the upper level indicates that zinc overload occurs readily. The body compensates by excreting more zinc.

12) Lines 355-357 : This sentence is vague and not needed. Merge with the earlier sentence (353-355). Also, dahi is yogurt. So you can combine references 95 and 98.

Reply: Revised, the dahi citation is now included with studies on yogurt.

13) Lines 365-367: The information provided is not true. Clarify that not all dairy proteins promote zinc absorption. 

Reply: See references 8 and 99, and response to comment 5. Our statement that milk and protein are able to increase zinc absorption, is well supported by a body of scientific literature. Clarification of the individual protein types that make up various dietary protein sources, and exploration of evidence for their differential effects on zinc absorption, would be interesting, however, beyond the scope of this a review.

14) Lines 376-9: Include at-home strategies to degrade phytate (malting, fermentation, soaking, germination, etc.)

Reply: Reworded to, “Concurrent strategies to reduce phytate, (e.g., malting, fermentation, soaking, or germination) should also be developed where feasible.”

15) Lines 433: Comparison with iron deficiency seems out of place. Reformulate.

Reply: Moved this comparison up a paragraph to improve the logical flow.  

Reviewer 2 Report

Ms-nutrients-1894663 needs careful revision of the text for the presence of typos.

The review presented by the authors is detailed and in-depth in its various parts, although the text lacks the reference to the homeostatic values recommended both by the food intake tables and the optimal values of zinc in organic fluids such as exchangeable Zn, with reference to the different genres and at the age of individuals. The introductory part is however acceptable, as well as the part on the factors that influence the absorption and fortification of the ion.

The data reported on the various population surveys could be summarized in one or more tables, in order to have a more immediate overview of the data.

Noteworthy is the part relating to the strategies used for zinc fortification and absorption.

Author Response

Ms-nutrients-1894663 needs careful revision of the text for the presence of typos.

Reply: Thank you for your comments. We have carefully revised the text, and made sure there are no typos remaining.

The review presented by the authors is detailed and in-depth in its various parts, although the text lacks the reference to the homeostatic values recommended both by the food intake tables and the optimal values of zinc in organic fluids such as exchangeable Zn, with reference to the different genres and at the age of individuals. The introductory part is however acceptable, as well as the part on the factors that influence the absorption and fortification of the ion.

This would be a good follow-up, we recognize the need is there, and we will make it a priority for the future.

Reply: Homeostatic values recommended for zinc in food intake tables by age, sex, and life stage; and optimal values of zinc biomarkers, are complex topics. They are reviewed extensively in the respective references (15, and 66). Their inclusion would be beyond the scope of the current review.

The data reported on the various population surveys could be summarized in one or more tables, in order to have a more immediate overview of the data.

Reply: We appreciate this recommendation. We have added tables summarizing the various trends in zinc fortification:

Table 1: Comparison of zinc measures and relation to bioavailability

Table 2: Comparison of recent zinc fortification strategies

We have also added a figure outlining our recommendation for optimizing the development of future zinc fortification:

Figure 1: Optimizing zinc fortification efforts

Noteworthy is the part relating to the strategies used for zinc fortification and absorption.

Reply: Thank you.